# mGEODAR—A Mobile Radar System for Detection and Monitoring of Gravitational Mass-Movements

**DOI:** 10.3390/s20216373

**Published:** 2020-11-09

**Authors:** Anselm Köhler, Lai Bun Lok, Simon Felbermayr, Nial Peters, Paul V. Brennan, Jan-Thomas Fischer

**Affiliations:** 1Department of Natural Hazards, Austrian Research Centre for Forests (BFW), 6020 Innsbruck, Austria; s.felbermayr@mci4me.at (S.F.); jt.fischer@bfw.gv.at (J.-T.F.); 2WSL-Institute for Snow and Avalanche Research SLF, 7260 Davos Dorf, Switzerland; 3Department of Engineering, Lancaster University, Lancaster LA1 4YW, UK; l.lok@lancaster.ac.uk; 4Department of Mechatronic, Management Centre Innsbruck (MCI), 6020 Innsbruck, Austria; 5Department of Electronic and Electrical Engineering, University College London, London WC1E 6BT, UK; nial.peters@ucl.ac.uk (N.P.); p.brennan@ucl.ac.uk (P.V.B.)

**Keywords:** FMCW Radar, natural hazards, geophysical mass-movements, flow dynamics, movement detection

## Abstract

Radar measurements of gravitational mass-movements like snow avalanches have become increasingly important for scientific flow observations, real-time detection and monitoring. Independence of visibility is a main advantage for rapid and reliable detection of those events, and achievable high-resolution imaging proves invaluable for scientific measurements of the complete flow evolution. Existing radar systems are made for either detection with low-resolution or they are large devices and permanently installed at test-sites. We present mGEODAR, a mobile FMCW (frequency modulated continuous wave) radar system for high-resolution measurements and low-resolution gravitational mass-movement detection and monitoring purposes due to a versatile frequency generation scheme. We optimize the performance of different frequency settings with loop cable measurements and show the freespace range sensitivity with data of a car as moving point source. About 15 dB signal-to-noise ratio is achieved for the cable test and about 5 dB or 10 dB for the car in detection and research mode, respectively. By combining continuous recording in the low resolution detection mode with real-time triggering of the high resolution research mode, we expect that mGEODAR enables autonomous measurement campaigns for infrastructure safety and mass-movement research purposes in rapid response to changing weather and snow conditions.

## 1. Introduction

Gravitational mass-movements are a major hazard to infrastructure, inhabitants and tourists in mountainous regions. For snow avalanches in particular, authorities deal with the hazard by temporary or technical mitigation measures, for example timed closures of roads and ski resorts or the protection of infrastructure with long term mitigation measures like dams and galleries. However, real-time detection of avalanches is advantageous as it allows safety measures to be implemented only when needed, for example closing a road using traffic lights when mass movements like snow avalanches are detected on the mountain flank above. Currently, such real-time detection uses seismic or infrasound sensors in close proximity to the avalanche flow or radar technology that can observe the avalanche path from a stand-off distance.

The first avalanche radars were of continuous wave type that measured speed utilizing the Doppler effect and a high spatial resolution was achieved by pointing a narrow beam to certain locations of the slope [1]. Modern radars for avalanche detection also utilize the Doppler effect, furthermore they send pulses that estimate the range by time of flight analysis. Those pulse-Doppler radars can achieve a range resolution on the 10 m scale, and within these range gates a complete velocity spectrum up to 100 m/s is measured [2]. While such pulse-Doppler radars are effective for movement detection, they lack high spatial resolution, limiting their application for detailed scientific studies on the flow dynamics.

To increase the range resolution, a second generation of gravitational mass-movement radars were developed based on the FMCW (frequency-modulated continuous wave) principle. The first FMCW radar for snow movement was developed in 2010 and initially tested on a small-scale snow chute [3], and later installed in the Swiss full-scale avalanche test-site at Vallée de la Sionne [4]. This radar is known as GEODAR (GEOphysical flow dynamics raDAR) and achieves a range resolution of 0.75
m at around 100 Hz pulse repetition. Although FMCW radars can resolve a Doppler velocity directly, avalanches act as a distributed target, making the velocity processing complicated or even impossible. Nevertheless, due to the high spatial and temporal resolution of GEODAR, features in the flow can be tracked in time and space and their velocity indirectly estimated [5]. GEODAR is the main sensor of several scientific studies that significantly improve the current understanding of avalanche dynamics, namely the identification and characterization of flow regimes within the avalanche [6].

GEODAR has been constantly modified and improved in the following years [7]. A major design change was introduced with the mark IV system when the laboratory waveform generator to produce the FMCW triangular wave was exchanged with a more compact and reliable direct digital synthesiser (DDS) [8]. Another capability of GEODAR is the phased-array receiver to localize the avalanche in cross-range, however, the large width of the avalanche made the array processing difficult [9]. In the mark V system, the receiver array was replaced with several directional antennas, each pointing at a slightly different cross-range angle. A disadvantage of the modular GEODAR system is its large size and measurements have only ever been taken from the slope of the Vallée de la Sionne test-site.

In this paper, we present a portable FMCW radar which works in a similar fashion to GEODAR mark IV but consists of only one receiver channel. The main idea is to keep the radar compact and reconfigurable so that measurement campaigns can be performed quickly in response to weather and snow conditions at various locations. Therefore the radar must not only be portable but also requires a battery-based power supply. Furthermore, the radar needs the possibility to trigger a measurement on its own, since an external signal of the avalanche start is usually not provided. This leads to the development of a low-resolution detection mode to perform continuous data recording and real-time detection processing.

Unfortunately, no data of a real mass-movement event could be acquired throughout the first year of operation, but we demonstrate the radar performance with freespace range data from a moving car. Additional loop cable test measurements show that the flexible high-frequency design of the hardware enables users to optimally choose settings of the frequency generation. We maximize the signal strength of the chirp pulses for the high-resolution research mode and low-resolution detection mode. We present a simple optimization method that utilizes only the hardware components and acquisition system itself and does not rely on any costly and sensitive high-frequency laboratory equipment. Thus, the signal generation settings can be optimized directly in the field.

The radar hardware design and the signal generation together with the FMCW data processing are presented in Section 2. Radar performance measurements and optimizations are shown in Section 3. The results and implications for further radar developments and for gravitational mass-movement measurements in general are discussed and concluded in Section 4.

## 2. Materials and Methods

The mGEODAR (mobile GEODAR) is an X-band FMCW-type radar with a centre carrier frequency of 10.4
GHz and 400 MHz bandwidth. The RF circuitry of mGEODAR is based on a previously developed radar EREDAR (Erebus radar), which is a purpose-built radar for monitoring the lava lake level in the Erebus volcano, Antarctica [10]. Although the system requirements to capture the variations of a lava lake in the range of centimetres per minute are quite different compared to the rapid movements of gravitational mass-movements, the digital nature of the signal generation enables the flexibility for various usage. Most of the radar hardware descriptions resemble EREDAR [10] and are only summarised here. First, we describe the radar hardware with settings of the current system featuring transmit frequencies from 10.2 GHz to 10.6 GHz. In the following subsections, we describe common FMCW design variables and focus on the flexibility of the radar hardware in respect to a variable signal generation and the baseband filter.

### 2.1. Radar Hardware

An overview of the mGEODAR hardware is shown in Figure 1. A key component in the signal generation chain is the DDS (AD9914, no. 1) on the transmitter board, which is clocked at $f_{s}=3.48\;G\mathrm{Hz}$ with a variable oscillator (AD4351, no. 2). The DDS divides this clock signal by 24 and allows for integer arithmetic for all the frequency calculations. The DDS generates a linear frequency ramp, hereafter called chirp signal. The chirp signal contains the fundamental frequencies from 0 to $f_{s}/2$, but also contains additional higher order harmonics called Nyquist-images [11]. A band-pass filter (Figure 2A) is used to select the image that gives the desired frequency ramp $f_{ch}$ of 400 MHz bandwidth at centre frequencies of around 2.4
GHz in the first image. This chirp signal $f_{ch}$ is up-converted with a local oscillator $f_{lo}$ (ADF5355, no. 3) to yield the desired transmit frequency $f_{tx}$ of 10.2 GHz to 10.6 GHz (Figure 1, annot. C).

The power amplifier (AM31-10.2-10.7-37-37) with 37 dB gain is in between the transmitter board and the transmit antenna. To keep the radiation pattern focused towards the avalanche track, the antenna specifications are chosen with a narrow azimuth angle and wider elevation angle. Two sector antennas (SA15-90-104V-D1 from Cobham) with 15 dB gain are used for transmit and receive purpose. Their beamwidth is around 85${}^{\circ }$ azimuth and 7${}^{\circ }$ elevation, but we mount the antenna with 90${}^{\circ }$ rotation to capture a complete vertical slice of the flow path.

On the receiver board, the received signal is first conditioned with a series of low-noise amplifiers and a cascade of high- and low-pass filters. Then the signal is down-converted with the local oscillator frequency $f_{lo}$ and the corresponding chirp signal $f_{ch}$, both taken from signal splitters on the transmitter board. The transmit and receive chain contain digitally-programmable amplifiers and attenuators to match the signal levels prior to all mixing stages. All the programmable components are controlled with a micro controller (VM2 from Venom Control Systems, Cambridge) using I2C and SPI serial communication protocols.

The resulting low-frequency deramp signal is passed through a custom built variable gain analogue baseband filter. The baseband filter characteristics are described in Section 2.4. Finally, the deramp signal is recorded using a 16-bit analog-to-digital converter (USB-6361 from National Instruments) with a maximum single-channel sampling rate of 2 MS/s. Since a binary marker signal indicating the beginning of a pulse is required as well, the sample rate reduces to 0.5
MS/s for a two channel acquisition to ensure multiplexing between the channels. The data is stored on a miniature computer with an Intel i7 CPU, 16 GB RAM and SSD storage. All digital signal processing (Section 2.5) is executed remotely on this computer, so that large data need not be transferred but rather the results are send as image files. For remote control and transfer of processed data an Teltonika RUT240 UMTS modem is installed that features an external antenna port. The radar hardware settings, acquisition and processing software can be remotely accessed and controlled with a Telegram messenger Bot interface implemented with Python.

### 2.2. FMCW Design Variables

A linear pulse in an FMCW radar is given by the bandwidth *B* and the pulse duration τ, or the so-called chirp rate $\alpha =\frac{B}{\tau }$. The target distance *r* is then a linear relation between this chirp rate, the signal propagation velocity *c* and the frequency *f* of the deramp signal
(1)$$r=\frac{c\;\tau }{2\;B}\;\cdot \;f=\frac{c}{2\;\alpha }\;\cdot \;f\;.$$

The range resolution Δr is the distance between two discernible targets and is related to the bandwidth *B* by $\frac{c}{2\;B}$. Approximating *c* with the speed of light gives Δr=0.375 m for the above mentioned mGEODAR settings. As avalanches are usually distributed targets with reflection returning continuously from the complete flow lengths, the range resolution equals the spatial sampling or so-called range gates along the mountain slope [5].

The maximum unambiguous range for an FMCW radar depends mainly on the largest resolvable frequency of the analog-to-digital converter (ADC), e.g., half the sample rate $f_{adc}$ gives a maximum range of r=937 m for the given system that can be adopted to longer ranges easily. Furthermore, the transmitted signal strength must be sufficient as it decays with $r^{-4}$ for point targets and $r^{-3}$ for distributed targets like avalanches [9]. Additionally for long range radars, the pulse duration τ needs to be longer than the two-way propagation time $\frac{2\;r}{c}$. The temporal resolution Δt depends on the pulse repetition rate that is limited by the pulse duration τ. When pulses are transmitted continuously Δt equals τ. Averaging over consecutive pulses for signal-to-noise improvements or sending pulse trains with different chirp rates reduces Δt accordingly.

A further constraint on the pulse parameters *B* and τ is imposed by the expected velocities *v* of the target. The target should not move between two range gates during a single pulse, that is, the maximal velocity should by smaller then $\frac{\Delta r}{\tau }$, otherwise the frequency of the deramp signal is smeared between several range gates similar to effects caused by pulse non-linearities [12]. Additionally, a moving object with radial velocity *v* causes a Doppler shift proportional to the speed and wavelength λ. As Doppler processing is difficult for distributed targets, this frequency shift $\frac{2\;v}{\lambda }$ manifests as a ranging error in Equation (Equation 1). A target approaching with 20 m/s imposes a ranging error of 0.52
m.

The wavelength λ of the transmitted signal is 0.03
m (10 GHz) and resembles the minimal size of objects that cause reflections rather than scattering. A powder cloud consists of single millimeter sized snow crystals and is therefore transparent to X-Band signals. Significant reflections are received from the flowing dense avalanche underneath the cloud [1]. Furthermore, X-Band radar permits avalanche observations independent of weather and visibility conditions.

### 2.3. Reconfigurable Signal Generation

The mGEODAR system comprises of two oscillators and a DDS as signal generation components that can be configured individually to give a variable range of output signals (no. 1–3 in Figure 1). The limiting factor for the signal variability are the hardware components such as filters, signal splitter and mixers. Furthermore, the external power amplifier and antennas also limit the usable range of frequencies, but since they can be easily exchanged, they are not taken into consideration here.

The high-frequency hardware is divided into three areas with each working at a distinct frequency range (annot. A–C in Figure 1). Valid ranges for possible frequencies are derived from the corresponding components and filter responses in each area (A) to (C). The filter transmission responses are shown in Figure 2. The three areas and their frequency ranges are namely the chirp signal $f_{ch}$ (1.9 GHz to 2.8 GHz), the local oscillator frequency $f_{lo}$ (7.6 GHz to 8.4 GHz) and the transmitted signal $f_{tx}$ (9.1 GHz to 11 GHz). Note, the values are approximate and are optimised later in Section 3.

The three areas are related to each other through the signal mixers, which perform an up-conversion of $f_{ch}$ by $f_{lo}$ on the transmitter side, i.e., $f_{tx}=f_{ch}+f_{lo}$. So not all combinations of $f_{ch}$ and $f_{lo}$ yield a usable transmit frequency $f_{tx}$ as for example the minimal achievable frequency is 9.5
GHz whereas the maximal frequency goes beyond the passband of (C) in Figure 2.

Beside the absolute frequency of the transmitted signal, the pulse bandwidth *B* is of greatest interest as it defines the range resolution Δr (see Section 2.2). A maximal pulse bandwidth of approximately 1400 MHz is permitted by the DDS, i.e., the sampling clock of the DDS is $f_{s}=3.5\;\mathrm{GHz}$ and the first Nyquist image ranges from $f_{s}/2$ to $f_{s}$, thus conservatively from 1.9 GHz to 3.3 GHz (see Figure 5A for the usable signal). However, the signal strength generated by the DDS in super-Nyquist operation follows a sinc-response, thus its output amplitude is far from flat over such large bandwidth [13]. A desired amplitude flatness can be slightly recovered by the high-pass filter response in the chirp signal area (Figure 2A). By choosing a different local oscillator frequency $f_{lo}$, the output of the DDS can be shifted to be compensated by the filter. Exactly such shifting of $f_{lo}$ is performed as signal optimization measurements in Section 3.

In this paper, the mGEODAR radar features two measuring modes of different linear frequency chirps which are referred to as research mode and detection mode. For the high-resolution research mode, a bandwidth of 400 MHz ranging from 10.2 GHz to 10.6 GHz is chosen. As discussed later, this resembles the maximal useful bandwidth with the mGEODAR radar system because the external power amplifier and antennas are designed for these frequencies. The research mode consists of a triangular modulation that varies continuously between up and down chirp once it is initiated. The chirp duration is τ=0.01s giving a pulse repetition rate of 100 Hz.

Clearly, the range resolution for gravitational mass-movement research purposes should be maximized, but the data rate is approximately 5 MB/s. This exceeds capability for storing a continuous acquisition and may even slow down real-time processing from a circular buffer for automatic movement detection. Therefore, a low-resolution detection mode is designed that complies with the national frequency regulations for radar usage of natural hazard monitoring in Austria [14]. These constraints limit a suitable chirp pulse for continuous transmission to a bandwidth of 30 MHz from 10.41 GHz to 10.44 GHz. For this mode, the DDS is programmed to generate a sawtooth modulation. The chirp pulses are not transmitted continuously but are triggered with an external logic pulse. Currently, the detection mode consists of a burst of 25 pulses with each a duration of $\tau =7.5\times 10^{-4}\;s$ and a 10 Hz repetition of these bursts. These 25 short pulses are averaged during offline processing to increase the signal-to-noise ratio.

The frequency flexibility that the filters in mGEODAR permit can also be a disadvantage as any external signal is received as interference rather than rejected. For example, if a second radar is running simultaneously for comparison and validation purposes, it is usually insufficient to tune both radar outputs to separate frequency bands because they still may interfere. A second radar must therefore operate at frequencies outside the above stated valid frequencies for $f_{tx}$ (panel C in Figure 2). However, external narrow bandpass filters may connect between the receiving antenna and the receiver unit. Such external filter is, for example, used in the GEODAR receiver chain to prevent saturation and signal leakage from a nearby pulse Doppler radar that operates at a similar frequency range [9].

### 2.4. Baseband Filter

A great advantage of FMCW radars is that range gain control can be achieved with active filtering at baseband frequencies [15]. The baseband filter on the deramp signal has three main functions. Firstly to suppress aliasing from higher frequencies. Secondly to remove any very low frequency content from direct coupling between transmit and receive antenna. Thirdly to ensure a range compensation of the signal strength as it typically declines with $r^{-3}$ of the target range *r*.

Note that with the above mentioned acquisition system, the first purpose of alias suppression is not well achieved with this filter. However, this issue is circumvented by a reasonable geometric arrangement of the radar and addition of a simple low-pass filter before the ADC. Usually, a mass-movement radar is installed on the valley base and the transmit/receive antennas point up towards the release zone at the mountain crest. Beyond the crest is usually free space and no targets of interest. The line-of-sight distance between the radar and the crest must therefore be smaller than the maximal unambiguous range.

The baseband filter is implemented as four stage active bandpass filter. The peak gain frequency is at approximately 430 kHz or 1600 m range. The variable gain settings for each stage permit digital selection of an active gain from 13 dB up to 57 dB. Each stage has a different frequency response and the cascading achieves a strong dB per decade roll-off. The simulated response of the baseband filter in mGEODAR is given in Figure 3 with the low gain response in the left panel (A) and the high gain response in the right panel (B).

The zero dB crossing of the filter is at 5380 Hz. The frequency of the deramp signal *f* for a target at range *r* is given by Equation (Equation 1), and in the current system with τ=0.01 s equals to a frequency rate of 267 Hz/m. Thus, the zero dB crossing lies at a range of 20 m. By adjusting the chirp duration τ, the frequency of the deramp signal *f* can be shifted into another region of the baseband filter. For example, a road in front of the flow path at 100 m can be practically blanked out at 0 dB of the hardware filter with the judicious choice of τ=0.00375 s. Additionally, another custom filter stage may be added in between the receiver board and the ADC that blanks out certain unwanted ranges.

### 2.5. Digital Signal Processing

The raw data stream from the ADC consists of two channels which contain the deramp time-domain signal and a marker pulse signal. The marker pulse is used to split the deramp signal into each chirp pulse waveforms, thus the time-domain signal is arranged into an 3D array with the axes hold the fast-time (time along the chirp pulse), the slow-time (centre time of all chirp pulses) and the chirp type (up- and down-chirp). Ash et al. [9] covers this splitting and reshaping process in detail. A time-domain example of the deramp signal is shown in Figure 4A,C. A spectral analysis with the fast-Fourier transform converts the fast-time into range for each chirp pulse according to Equation (Equation 1). The magnitude of the amplitude spectrum for a loop cable measurement is shown in Figure 4B,D. Note, any frequency shift due to the Doppler velocity of a target is neglected here due to the problems arising from distributed targets. A cosine-tapered window function is applied prior to the range analysis to taper each side of the chirp pulse to zero. Some spurious signals appear as discrete non-coherent regions in the time-domain data which can be notched out with cosine tapering as well. Such spurious signals are most likely frequency leakage in the hardware or external in-band signals, and such digital windowing has been successfully applied in the previous GEODAR radar processing chain (see Köhler [16] for an in-depth description of the complete data processing). Part of the optimization in Section 3 is to prevent signal degradation due to these spurious signals. For the detection mode, the spectral data of the 25 pulses in one burst are averaged in order to increase the signal-to-noise ratio.

The slow-time processing, also known as moving target identification (MTI), compares the response of several consecutive chirp pulses. A high-pass filter along the slow-time axis for each range bin enhances changes between the pulses while suppressing static regions. The exact filter properties depend on the target, for avalanches usually a large passband is found optimal [5]. However, very distinct flow patterns and flow regimes in avalanches could be enhanced by using narrow bandpass filters [17]. The same holds true for discrete point targets and here the MTI filter is a digital finite impulse response bandpass filter with normalized cut-off frequencies of [0.025,0.4] and a length of 151 pulses. The lower cut-off basically removes levelling artefacts from range normalization, while higher cut-off removes high-frequency noise such as pulse-to-pulse variations.

## 3. Measurements and Results

The radar performance and frequency configurations are tested with a coaxial cable loop setup connected between the transmitter and receiver board. The deramped signal intensity of a 90 m long loop cable is maximized by varying the local oscillator frequency $f_{lo}$ for the 400 MHz research mode and the 30 MHz detection mode. We propose that such an optimization is helpful for each new installation location of the radar hardware. Finally, a freespace range measurement of a moving point target shows the real application performance.

### 3.1. Signal Strength Optimization

To find optimal settings for the chirp signal $f_{ch}$ and the local oscillator $f_{lo}$, we vary the settings of the DDS and the local oscillator according to the desired transmit frequencies ($f_{tx}=f_{ch}+f_{lo}$). As mentioned before, the frequency band for our research mode is 10.2 GHz to 10.6 GHz and for the detection mode is 10.41 GHz to 10.44 GHz.

The measurement for signal strength optimizations were done with a 90 m long coaxial cable (S_04272_B from Huber&Suhner). This cable has a signal velocity reduction factor of 0.82, so the deramp signal peak is expected at around 55 m range in the radar data (Equation (Equation 1) for two-way travel). The cable has an attenuation of 1.073 dB/m at 10 GHz giving around 96 dB attenuation for the complete cable. We include the external 5 W power amplifier into the loop measurement to compensate for the high cable attenuation. The amplifier and attenuator on the transmitter board were set to not exceed the required input signal for the power amplifier of 10 dB, namely, the TX amplifier set to 27.9 dB and the TX attenuator to −4.5 dB. On the receiver board, the two variable amplifiers are set to 16 dB and the attenuator to 0 dB. All stages of the baseband filter were set to their high gain settings.

The most limiting factor for the usable signal is the filtering of the chirp signal (Figure 2A) which suppresses frequency components beyond the first Nyquist image of the DDS fundamental frequencies. The width of this first image ranges conservatively from 1.9 GHz to 3.3 GHz. However, the output signal strength of the DDS follow a *sinc*-function, thus we expect decrease of signal with an increase of the chirp frequency $f_{ch}$ [11].

We observe some spurious signals in the deramp signal. The instantaneous DDS frequency of these spurious is exactly a third of the set $f_{lo}$ frequency, and they correspond to expected spurs from the mixer. Thus, for the mGEODAR frequency range, the spurs only exist in-band for $f_{lo}$ between 7.68
GHz and 7.92
GHz, otherwise $f_{lo}/3$ is outside the frequency of the deramp signal.

We optimize the signal intensities of both measuring modes with the loop cable. For each $f_{lo}$ we took a separate 1 s loop cable measurement with all settings $f_{lo}$, $f_{ch}$ and chirp rate α=40 MHz/s controlled with the Venom micro controller. All the test measurements are done for both measuring modes individually and indicated with subscripts 400 and 30 for the research and detection mode, respectively. If a spurious signal lies within the deramp signal, a custom cosine-tapered window is applied to the time-domain deramp signal to notch out the spike prior to the spectral analysis.

We aim to maximize the magnitude of the Fourier transformed deramp signal by varying $f_{lo}$ at the range of the cable length. For example, Figure 4B,D show the frequency spectrum of the research and detection mode, respectively. The peak at the cable length is clearly visible for these measurements at $f_{lo}=8.12\;\mathrm{GHz}$, but its strength varies for other frequency settings. Figure 5A shows the intensity *P* at the apparent cable length (55 m radar range) and the error bars of *P* extend to each side for one standard deviation that is derived from approximately 100 pulses in the 1 s measurement. The standard deviation is derived with Equation (2) in Taraldsen et al. [18] for conversion into decibel scale.

Additionally, the intensity *B* of the background signal for both measuring modes is shown. There are some unwanted spectral components at for example 470 m and 580 m (see Figure 4B,D). We define the background signal *B* in a conservative way, to take these unwanted spectral components into account. We define the background signal *B* as the largest of those peaks. And again a standard deviation is used for error bars of *B*, but this time refers to the variation of the background spectrum across range. Thus, the larger the error bars, the greater is the background signal variation and the larger are the unwanted spectral components in comparison to the average spectrum magnitude.

The curves for the intensities *P* and *B* of both measuring modes in Figure 5A have a similar shape with a signal drop around $f_{lo}=8\;\mathrm{GHz}$ as well as below 7.8
GHz and above 8.2
GHz. The signal of the loop cable target *P* maximizes at $f_{lo}=8.12\;\mathrm{GHz}$. The largest error bars for the background of the spectrum are found for the smaller values of $f_{lo}$, which correlates with the occurrence of the spurs in the deramp signal at instantaneous chirp frequencies $f_{ch}=f_{lo}/3$.

Beside the absolute intensities, the signal-to-noise ratio (SNR) is of similar importance. We define the SNR as the ratio between the cable intensity *P* and the background intensity *B*, and is shown in Figure 5B. The SNR follows roughly the same trend as for the individual intensities *P* and *B*. Even though the SNR is highest for around $f_{lo}=7.9\;\mathrm{GHz}$, the spectra are more disturbed for this settings as shown with the larger error bars. Again the optimal setting found for $f_{lo}$ is 8.12
GHz.

The deramp signal and the corresponding spectrum for the optimal frequency setting $f_{lo}=8.12\;\mathrm{GHz}$ is the content of Figure 4. The top row shows the data for the research mode and the bottom row shows the detection mode. The zoom in panel (A) focus on the area of the deramp signal that has the same extend as the deramp signal for the detection mode in panel (B). It is easy to see, that the deramp signal *f* is very similar in both plots as 11 periods are clearly visible and the signal strength is comparable with around ±0.25 V.

The range resolution for both measuring modes manifests in the spacing of the points in the frequency spectrum. For the detection mode, the peak of the cable measurement spreads across neighbouring value to about 10 m. For the research mode, the sampling of the spectrum is much narrower and the cable peak significantly spreads around 1.5
m. Additionally, the spike-like unwanted spectral components in the frequency spectrum are more pronounced in the research mode measurements and only the strongest ones are found also in the detection mode due to the lower frequency sampling. We believe these unwanted spectral components are caused from internal electronic noise of the radar components as their position in the spectrum are independent on any frequency setting $f_{lo}$ or $f_{ch}$. Being aware of the disturbed ranges enables to discard them for any further data processing.

### 3.2. Moving Point Target

A freespace range radar measurement was performed along a straight road with a moving car as target. Both radar antennas were set up approximately 1.5
m above ground with a spacing of 4 m. The external 5 W power amplifier was used for these measurements in between the transmitter board and the transmit antenna. The frequency $f_{lo}$ was set to 8 GHz instead of the found optimum from Section 3.1, and thus the signal-to-noise ratio is expected to be lower than the optimum.

Figure 6 shows the range-time diagrams of a car driving away from the radar location at range 0 m. The slope of the streak signature in the range-time diagram gives the velocity of the car. The car accelerates for the first 100 m, and reaches a constant velocity of approximately 15 m/s. Note, both panels result from two different measurements as the research mode (A) and the detection mode (B) cannot be measured simultaneously.

The colour intensity is a moving target identification (MTI) based on the digital filter described in Section 2.5. The values are displayed in decibel referenced to a measurement of the same scene without any movement, thus, non-moving regions in the MTI are ideally zero. However, speckle-like noise appears especially for the research mode (Figure 6A) that could be dealt with by averaging across single chirp pulses. For the detection mode (Figure 6B), such averaging is already done along the 25 short pulses during one burst (see Section 2.3). The SNR in the MTI image of the car compared to the background is approximately 10 dB for the research mode and 5 dB for the detection mode. The moving car gives a clear discrete target in the MTI data that exceeds the speckle-like background noise at least for ranges below 200 m.

The temporal and spatial resolution of both measuring modes is shown in the insets of Panel A and B that both span approximately over 20 m and 1.5
s. The car in the research mode measurement in panel A spans several range bins with each a spatial resolution of Δr=0.375 m. However, the detection mode in panel B has a resolution of Δr=5 m and the car appears in a single range bin at each time.

The effect of the active baseband filter (Section 2.4) is clearly visible at ranges below 20 m. To prevent any cross-talk and direct coupling between the transmit and receive antennas, the baseband filter strongly suppresses all low frequencies in the deramp signal. Furthermore, the signal intensity of the car drops slightly between a range of 100 m and 200 m. The geometric attenuation is only partly compensated by the filter. For ranges larger than 200 m the signal intensity diminishes further due to additional attenuation of a disturbed Fresnel zone. The Fresnel ellipsoid touches the ground as the antenna were only slightly raised above ground. This is usually not a problem with measurements in the mountains as the antennas are mounted high on the opposite slope of the flow path rather than measuring complete parallel to the ground.

## 4. Discussion

Wide bandwidth radars for high-resolution gravitational mass-movement monitoring like the mGEODAR, and also its predecessor EREDAR, are challenging to construct as matching of wideband components and in-band spurious signals limit their performance. For example, the optimal frequency settings for the mGEODAR radar are slightly different from what to expect from the filter responses in Figure 2A. And some unwanted components in the frequency spectrum of the deramp signal appear as ghost targets (Figure 4B). Thus, the performance of the complete system and the interplay of the components need to be verified and optimized.

Usually optimizations of high-frequency systems require costly laboratory test equipment. However, due to the integrated design of the radar boards, they cannot be used in intermediate stages between most of the components. Thus, we directly used the capabilities of the radar itself to find frequency settings that optimize the received signal strength. This approach has the advantage of including the whole signal chain in the optimization process.

A similar optimization should be performed for each installation locality as for example the antenna cross-talk depends on the installation site and its geometry. Instead of a loop cable, such measurements could be done with active targets at several distances in the radar field of view. By knowing the geometric arrangement of the radar installation, potential cable lengths and the location of the active target with GPS measurements, the signal strength optimization procedure can done before any experiment. An active target may simply consist of two antennas with an amplifying component in between [19]. These targets prove also useful for monitoring the radar performance throughout longer time spans like the complete winter season.

For the current mGEODAR system we find an optimal LO frequency setting for the 400 MHz research mode at $f_{lo}=8.12\;\mathrm{GHz}$. Then the chirp signal $f_{ch}$ that is generated in the first Nyquist image of the DDS covers frequencies between 2.08
GHz and 2.48
GHz. The signal strength of a super-Nyquist output generally follows a *sinc*-function although gain flatness is desired [13]. We postulate using a slightly higher frequency for the chirp signal and a slightly lower frequency for the LO would be better and could avoid more low frequency attenuation of the HFCN-2275 (Figure 2A). However, this rejection filter actually equalizes the gain and thus compensates the decrease in signal strength towards the higher frequencies of the chirp signal. For the 30 MHz bandwidth detection mode the gain flatness is of lower importance, however, single spurs may have stronger effects on the signal degradation. We find an optimal LO frequency setting at $f_{lo}=8.12\;\mathrm{GHz}$ also for the detection mode.

Another possibility to increase the gain flatness would be to change the clock frequency $f_{s}$ of the DDS. A slight increase of $f_{s}$ towards 3.9
GHz may already shift the chirp signal $f_{ch}$ into a flatter region. Even though over-clocking of the DDS chip has been successfully tested [20], the chip may overheat more quickly in continuous monitoring usage and may cause erratic behaviour.

The current radar system can still improve on the pulse coherency in the research mode as the standard deviation between 100 pulses is relative high (Figure 4A). As a result, the background noise level of MTI image in the range-time-diagram (Figure 6A) sometimes shows strong speckles. Averaging of several pulses for the research mode would strongly decrease the temporal resolution of the radar. And the rapid movement of an avalanche may cause a smearing of details across range bins within the flow when averaged. We think the pulse coherency is degraded by the continuous triangular ramping that is based on the DDS internal timing settings and leave the initial phase of the chirp uncontrolled. As Long and Reschovsky [20] report, a digital triggering of a ramp start can improve the phase stability greatly. The EREDAR system utilizes digital triggering of the DDS and the acquisition system [10], and we will implement a similar chirp control for the research mode in a next step.

When this trigger impulse is generated by a digital output of the acquisition system, the binary marker that indicates the beginning of a pulse does not need to be captured. Instead, the acquisition system can be set to record only the deramp signal channel and the full recording speed of 2 MS/s is available. Furthermore, the data processing is accelerated as rearranging of the time-series data into single chirp pulses would be omitted. And the online data processing can request single chirp pulses or bursts of pulses as soon as the resources are free.

Up until now the data processing is performed offline and no trigger conditions to switch between the detection mode and research mode have been identified yet. The next step is to acquire real avalanche data, develop real-time data processing and motion detection algorithms. Since direct velocity estimation with FMCW technique is complicated and so far rather impossible for large distributed targets that constantly change the reflectance due to variable composition and deformable surface, any motion detection needs to compare the return signal between several pulses, thus, apply image identification and classification algorithms to the 2-dimensional range-time MTI data (Figure 6).

The signal intensity for the moving car is difficult to compare with the signal strength expected from an avalanche. The car is a point target and its scattering intensity is usually higher than the background, in contrast for avalanches, the background scattering of the snow cover is of similar magnitude to the moving snow. However, the previous FMCW radars GEODAR I to V have indeed enough sensitivity to successfully image very different avalanches that range from small loose snow avalanche to large powder snow avalanche and from dry to wet snow avalanches [17]. Still, the quantitative interpretation of the MTI signals of avalanches is very challenging due to a complicated interplay of the radar signal and the MTI filter response with the avalanche width, height and velocity, the internal flow turbulence, and snow type, granule size and surface roughness of the avalanche.

## 5. Conclusions

We present the hardware of a custom-build mobile FMCW radar system for the purpose of detecting, monitoring and measuring gravitationally driven geophysical mass-movements such as snow avalanches. We anticipate that very similar radars will be employed in the future to monitor other natural mass movements like debris flow, rock fall and also very rare pyroclastic density currents on volcano flanks. However, the development of such mass-movement detection radars definitively benefits from snow avalanche research as these flows can be triggered manually and occur comparably more frequent throughout the complete winter season. Thus, the hardware, software settings and real-time processing algorithms can be developed more easily for snow avalanches and later adopted to match the other mentioned flow types that are sporadic natural events and usually exhibit a complex transient signature in time and space. Indeed, more research is required in feature recognition algorithms on the range-time data to differentiate a mass movement event from other moving targets such as waving trees in the field of view.

For various measurement locations in rapid response to changing climate and snow conditions, the presented optimization procedure will prove useful to quickly find the optimal frequency settings for any new measurement location. We see a main advantage of the procedure, that basically sweeps through the full valid frequencies of the hardware, without the need for additional sensitive laboratory equipment. Employing solely the radar hardware and capability of the analog-digital converter ensures that the complete signal chain, the cabling and antenna orientations as well as potential external noise sources are optimally taken into account at each location.

Even though no natural avalanches could be recorded until now, the presented tests and results of a non-distributed moving target are promising, however, further evaluation on real events is needed. In the future, the data acquisition and processing will be further optimized for computational speed and efficiency. A next step will be the development of a hardware-based FMCW processing using FPGA technology rather than software-based offline processing which requires decent computing power and energy consumption. Beside the flexible usage cases for the radar, the overall costs of all mGEODAR hardware components are around 10 k Euro and will promote the wide use of radar technology in mass-movement and general environmental monitoring, detection and measurements across various disciplines and as ground-truthing for satellite based radar remote sensing of, for example, the snow cover properties.

## Figures and Tables

**Figure 1 sensors-20-06373-f001:**
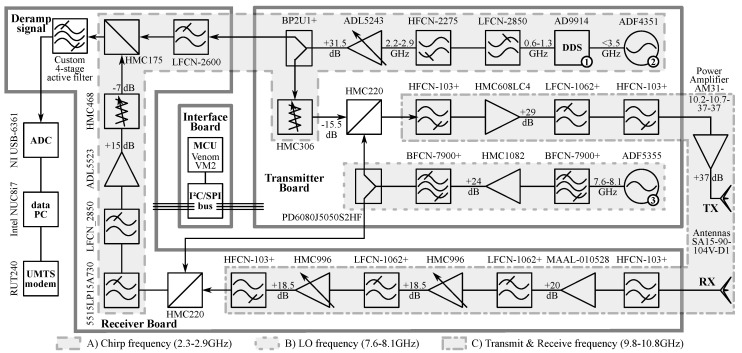
Hardware schematic with high-frequency components, data acquisition and controlling devices of radar mGEODAR. Three printed circuit boards (PCB) boards host the radar hardware: The transmitter board is shown on top-right, the receiver board is on the bottom-left and the interface board for control of HF components is in the middle. The data acquisition, storage and processing devices are shown on the far left. Signal generator and oscillators are enumerated (no. 1–3). Annotation with light gray boxes and letters (annot. A–C) indicate areas with same frequency. The frequency responses of all filters in these boxes are drawn in Figure 2.

**Figure 2 sensors-20-06373-f002:**
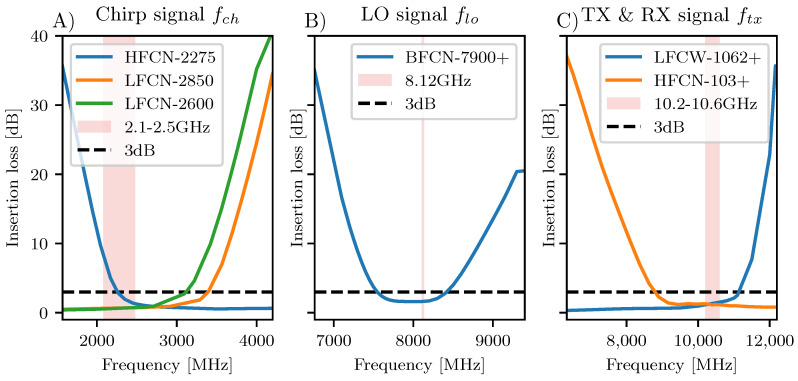
The insertion loss frequency responses of the HF-filter in mGEODAR. Each subplot gives the filter used in the frequency areas of annotated boxes (**A**–**C**) in Figure 1. Pink area denotes the frequency ranges of the optimized system (see Section 2). Note, the line plots are for single filters and they are cascaded so that their responses accumulate accordingly. Data taken from the manufacturer datasheets.

**Figure 3 sensors-20-06373-f003:**
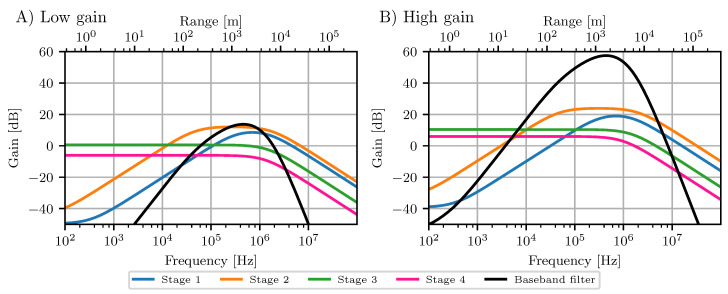
Amplitude response of the 4-stage active baseband filter in mGEODAR at low gain (**left A**) and high gain (**right B**) setting in black. Stage 1 and 2 define the bandpass shape and stage 3 and 4 cause a strong roll-off to high frequencies. The zero crossing in high gain setting is at 20 m and the maximum pass band ranges roughly from 800 m to 3000 m.

**Figure 4 sensors-20-06373-f004:**
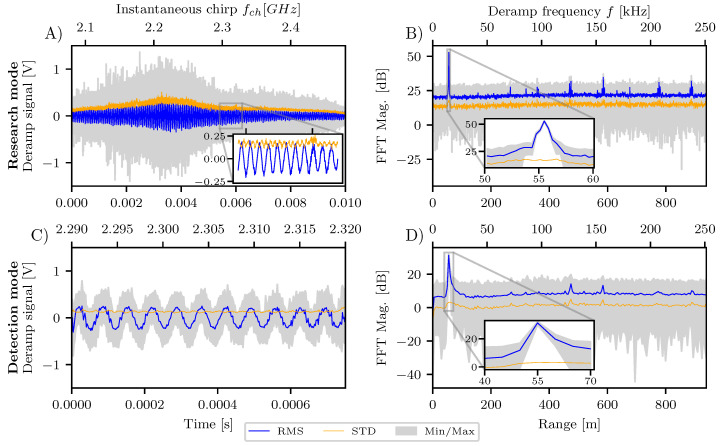
Deramp signal and spectrum statistics for both measuring modes with $f_{lo}=8.12\;\mathrm{GHz}$. The research mode measurement is shown in the top row (**A**,**B**), the detection mode in the bottom row (**C**,**D**). The left panels show the data acquired with the ADC with the pulse duration time on the bottom axis and the corresponding instantaneous chirp frequency of the DDS $f_{ch}$ on the top axis. The right panels are the amplitude spectrum of the deramp signal *f* with bottom axis converted to radar range.

**Figure 5 sensors-20-06373-f005:**
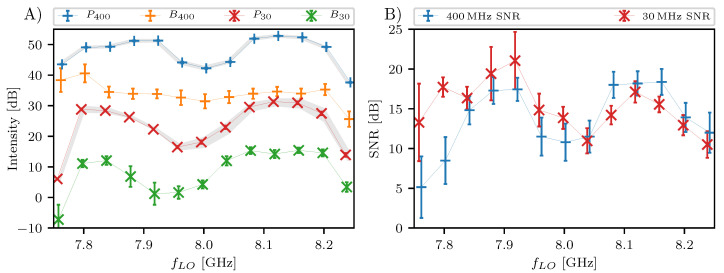
(**A**) Peak intensity of cable length (*P*) and averaged background (*B*) in a 90 m loop test setting for different $f_{lo}$ frequencies and both measuring modes. Error bars are two standard deviations wide. (**B**) Signal-to-noise ratio between cable peak *P* and background intensity *B* for the research mode with 400 MHz ramp (blue) and the detection mode with 30 MHz ramp (red).

**Figure 6 sensors-20-06373-f006:**
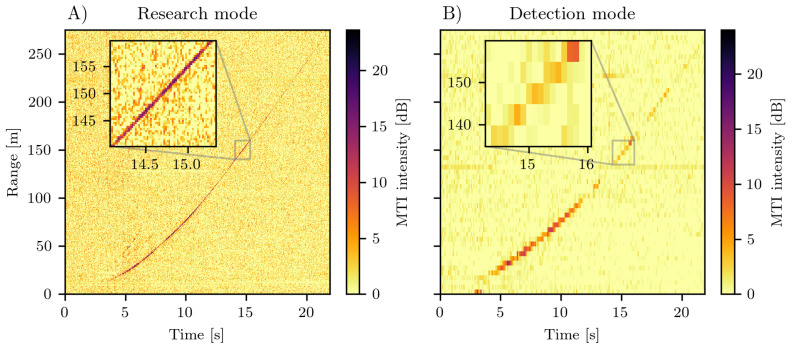
A car departs from the radar location at range 0 m. The colouring is the intensity of the moving target identification for (**A**) the high resolution research mode and (**B**) the low resolution detection mode. The insets show a zoom of approximate similar range and time extent, thus the difference in resolution of both measuring modes can be seen. Note, the panels show two different data sets as both modes cannot be measured simultaneously.

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
