# Peer review of "mGEODAR—A Mobile Radar System for Detection and Monitoring of Gravitational Mass-Movements"

_sensors, 2020, doi:10.3390/s20216373_

Round 1
Reviewer 1 Report
Review of the Manuscript (sensors-965761)
“mGEODAR – a mobile radar system for detection and monitoring of gravitational mass-movements”
submitted to Advances in Sensors by Anselm Kohler& Co-authors
General Comments
This paper reports on a new portable, compact, and fully reconfigurable radar system (mGEODAR) for detection and monitoring of gravitational mass movements. Gravitational mass-movements, like snow avalanches, are a major hazard in mountainous regions. The real-time detection of the avalanches, routinely done by means of seismic and infrasound sensors, is undoubtfully pivotal as it might allow safety measures to be implemented only when strictly needed. The main idea behind this project is to produce a portable FMCW (frequency-modulated continuous wave) radar system and a low-resolution detection mode. Main goal is performing continuous data recording and real-time detection processing so that measurement campaigns can be quickly carried out in response to rapidly changing weather and snow conditions at various locations.
Unfortunately, the authors weren’t able to test the new system with a real avalanche, but some satisfactory tests are done with a non-distributed moving target (a car). The actual challenge is the evaluation of the performances with real avalanches, which might be very distributed targets.
The subject is very interesting, even scientifically and technically sound and funded on rather convincing claims, which are satisfactory supported by the experimental data. Setting of the system is extensively discussed and justified based on state of the art.
Hereafter, I am going to suggest few changes the authors might consider for hopefully improving the readability of the paper.
I recommend publication of the manuscript after minor changes
I just draw your attention on the fact that there is a gap in the line numbering in the Manuscript, namely between the lines 124-125
Comment 1(Abstract) Routinely in the abstract the results have to be summarized. Please, try providing more clearly the main outcomes reported in the manuscript.
Comment 2 (4. Discussion and Conclusion). I suggest to split in two separate sections “4. Discussion” and “5. Conclusion”
Missing Line number (it should be 126): add “r” next “The target distance r…”
Figure 6: Is the moving target identification (MTI) an adimensional quantity? If any, please add units on the colour bar.
Reviewer 2 Report
This paper proposes mGEODAR based on GEODAR, and some suggestions is as following:
1. The loop cable is measured to verify the increase in signal. How can the distance to be measured on site increase the signal when the distance is not calibrated?
2. Two working modes for mGEODAR can not work at the same time. The detection mode is continuous data recording and real-time detection processing. What is the trigger condition of the research mode?
3. mGEODAR is designed for mass-movement detection, aiming at distributed targets. However, the source of the verification data for the hardware and signal processing method is a moving car. There are two issues that need to be considered: a. The car is a point target and its scattering intensity is significantly higher than the background, but for distributed targets such as avalanches, the background scattering intensity is similar to the move target, and the target signal extraction is more difficult. b. It is mentioned in the article that because the diameter of the small snow crystal particles is smaller than the wavelength, the scattering intensity can be ignored, but does the absorption effect of a certain number of snow crystals on electromagnetic waves need to be considered in the signal processing?
